# Absence of Wharton’s Jelly at the Abdominal Site of the Umbilical Cord Insertion. Rare Case Report and Review of the Literature

**DOI:** 10.3390/medicina57111268

**Published:** 2021-11-18

**Authors:** Radu Botezatu, Sandra Raduteanu, Anca Marina Ciobanu, Nicolae Gica, Gheorghe Peltecu, Anca Maria Panaitescu

**Affiliations:** 1Department of Obstetrics and Gynecology, Filantropia Clinical Hospital, 71117 Bucharest, Romania; radu.botezatu@umfcd.ro (R.B.); sandranedelea@gmail.com (S.R.); anca.ciobanu@umfcd.ro (A.M.C.); gheorghe.peltecu@umfcd.ro (G.P.); anca.panaitescu@umfcd.ro (A.M.P.); 2Department of Obstetrics and Gynecology, Carol Davila University of Medicine and Pharmacy, 71117 Bucharest, Romania

**Keywords:** Wharton’s Jelly absence, abnormal cardiotocography, umbilical cord

## Abstract

Wharton’s jelly is a specialized connective tissue surrounding and protecting umbilical cord vessels. In its absence, the vessels are exposed to the risk of compression or rupture. Because the condition is very rare and there are no available antepartum investigation methods for diagnosis, these cases are usually discovered after delivery, frequently after in utero fetal demise. We report the fortunate case of a 29-year-old nulliparous woman, with an uncomplicated pregnancy, admitted at 39 weeks in labor where a persistently abnormal cardiotocographic trace led to delivery by cesarean section of a healthy 3500 g newborn. After delivery, a Wharton’s jelly anomaly was identified at the abdominal umbilical insertion (umbilical cord vessels, approximately 1 cm in length, were completely uncovered by Wharton’s jelly), which required surgical thread elective ligation. In the presence of a persistently abnormal CTG trace, in a pregnancy with no clinical settings suggestive of either chronic or acute fetal hypoxemia, the absence of Wharton’s jelly should be taken into consideration in the differential diagnosis.

## 1. Introduction

The normal umbilical cord (UC) contains one umbilical vein which carries oxygenated blood and two umbilical arteries; the three vessels are surrounded throughout the whole length of the UC by a gelatinous tissue called Wharton’s jelly. The normal insertion of the UC is central or slightly eccentric, directly into the placental disk at one end and the umbilicus on the fetal abdominal wall at the other end. The average length of the umbilical cord at term is approximately 55 cm (increasing with advancing gestational age, from a mean of 32 cm at 20 weeks gestation to 60 cm at term) [1]. The average diameter and circumference of the UC in a normal term infant are 1.5 and 3.6 cm, respectively.

Wharton’s jelly is a specialized connective tissue composed of myofibroblasts and extracellular matrix—proteoglycans, glycosaminoglycans, and hyaluronic acid—with the primary function of protecting the umbilical blood vessels [2]. This structure neutralizes the external pressure influence on blood flow between the placenta and fetus, promoting adequate blood flow to the fetus in cases of umbilical cord compression during pregnancy or delivery [3].

After delivery, all placentas should undergo a basic examination including color, length of the umbilical cord, thickness, number of cord vessels and weight of the placental disk. Complete gross and histopathologic examination of the placenta should be obtained when clinically indicated by adverse maternal, fetal or neonatal outcomes [4]. Some findings have no clinical significance, but some structural changes are strongly associated with intrauterine growth restriction, fetal death, fetal distress in labor and increased incidence of cesarean delivery [5]. A thin umbilical cord is a result of a deficiency of Wharton’s jelly. The vessels in a thin cord are more vulnerable to compression, which may explain the association between thin umbilical cords and intrauterine growth restriction and abnormal intrapartum fetal heart rate tracings [6]. The loss of protection by Wharton’s jelly can lead to compromised fetoplacental circulation and subsequent fetal death [7]. The complete or segmental absence of Wharton’s jelly is a very rare abnormality, with only a few cases reported in the literature.

The aim of this paper is to report a case where the absence of Wharton’s jelly was diagnosed postnatally in the setting of an abnormal labor with persistent modified CTG traces. A literature review of published cases of the absence of Wharton’s jelly is included.

## 2. Case Report

We present the case of a 29-year-old nulliparous, 39-week pregnant woman who was hospitalized for irregular contractions in early labor, no ruptured membranes, no bleeding and very good fetal movements. The pregnancy follow-up was uneventful, with both first and second-trimester anomaly scans within normal limits. There was no other associated pathological medical history. Clinical examination and vital signs were normal. At the ultrasound examination, the placenta was high posterior, there was normal amniotic volume, cephalic presentation and active fetal movements, and estimated fetal growth of 3500 g with normal umbilical, cerebral and uterine arteries doppler.

During routine cardiotocography, a pathological trace was recorded—tachycardia and late deceleration (Figure 1). As per guidelines [8], immediate action to correct reversible causes and additional methods to evaluate fetal oxygenation were required. Because the patient requested a vaginal delivery and chronic hypoxia needed to be excluded, the artificial rupture of membranes was performed to rule out the presence of meconium and to assess other possible causes of fetal hypoxia. The amniotic fluid had a normal aspect, with no signs of meconium, blood or foul smell. Having excluded infection and chronic hypoxia, and because the trace was reassuring with stable baseline and normal variability, the plan was to reassess progress in 2 h under continuous CTG monitoring, IV hydration and maternal repositioning. Two hours later, another event raised the concern of suspected fetal compromise—a prolonged 4 min late deceleration immediately after a high amplitude fetal movement (Figure 2)

A decision was made for a category 2 cesarean section. A healthy 3500 g male was delivered in good condition (Apgar score 8). The macroscopic placental examination was normal. During the first routine clinical examination of the neonate, a Wharton’s jelly anomaly was identified at the abdominal umbilical cord insertion. For approximately 1 cm of length, umbilical cord vessels were completely uncovered by Wharton’s jelly. The condition required surgical thread elective ligation of exposed umbilical cord vessels (Figure 3).

## 3. Discussion

We present a case where the absence of Wharton’s jelly was observed after the delivery of a healthy term baby with abnormal acute CTG trace modifications during labor.

The absence of Wharton’s jelly is a rare event with only several cases reported in the literature. We conducted a review of the literature in the PubMed database using the search terms ‘Wharton’s’ or Wharton jelly’ and ‘absence’ or ‘absent’. There were sixty-one results retrieved from the beginning of indexing to 2021. We excluded manuscripts not presenting clinical cases. Ten cases were found that matched our search, published between 1985 and 2020. Only two cases reported neonates that were discharged in good condition, six cases were associated with perinatal death, one case was associated with a congenital malformation, a persistent vitellointestinal duct, and the other one had severe neonatal morbidity (Table 1).

Changes in the consistency of Wharton’s jelly affect umbilical cord blood flow and can contribute to intrauterine growth restriction (IUGR) and even fetal demise. The segmental absence of Wharton’s jelly is an extremely dangerous condition, as umbilical vessels are directly exposed to compression or damage. The absence of Wharton’s jelly can lead to fetal chronic hypoxia [5,10,14,15]. In our case, there were no signs of fetal hypoxemia before or after delivery. The acute modifications seen on the CTG trace recorded in our case could be explained by the fact that umbilical arteries and veins were suddenly severely compressed at the abdominal insertion portion during active fetal movements.

Normal placental function and fetal oxygenation are prenatally assessed by the ultrasound evaluation of fetal growth, Doppler evaluation of umbilical artery blood flow, the amount of amniotic fluid and maternal perception of fetal movements. Unfortunately, none of these markers were proven to be useful in predicting adverse outcomes related to abnormal Wharton’s jelly, as all the cases reported so far had a sudden occurrence, mostly in normally growing fetuses with normal amniotic fluid. In our case, the Doppler evaluation of the umbilical artery before delivery was normal. Abnormal CTG monitoring with decreased variability and persistent late decelerations was the only prenatal finding which indicated an emergency delivery, and this is in accordance with other reported cases [5,10,14].

Previously reported cases of the absence of Wharton’s jelly describe an association with either meconium-stained liquor [5] or oligohydramnios, fetal growth restriction, preterm delivery and perinatal death [3]. None of these findings were true in our case.

The prenatal evaluation of Wharton’s jelly is not part of routine practice, but modifications in its amount and composition have been linked to unfavorable outcomes. [16]. Most studies investigating the amount of Wharton’s jelly and its implications examined the size of the umbilical cord on a cross-sectional plane. Growth-restricted fetuses have a smaller cross-sectional area of the umbilical cord when compared with normally growing fetuses and these differences are related to the amount of Wharton’s jelly and umbilical vein size [17]. Moreover, some reports showed that the amount of Wharton’s jelly expressed as the umbilical cord cross-sectional area is correlated with neonatal birth weight [18].

The proposed theories in the pathogenesis of absent Wharton’s jelly are related to possible degeneration, early incomplete fusion of amniotic and mesenchymal umbilical tissue, or hypoplasia of amnion and secondary loss of Wharton’s jelly [9]. However, a clear explanation for this rare abnormality is not known.

In our case, the absence of Wharton’s jelly at the abdominal wall insertion could also meet the prerequisites for umbilical cord rupture. Therefore, when an abnormal CTG is associated with active fetal movements, a high grade of suspicion is required for this rare condition and a low threshold regarding the decision for delivery is needed.

## 4. Conclusions

The absence of Wharton’s jelly is one of the rarest conditions seen in the professional life of an obstetrician. The segmental absence of Wharton’s jelly is almost impossible to diagnose during prenatal ultrasound evaluations and most of the ultrasound markers used in assessing fetal well-being are not good predictors of fetal complications in cases associated with Wharton’s jelly abnormalities. Here we present a fortunate case, where persistent an abnormal CTG trace without any other explanation announced this rare finding—the absence of the Wharton’s jelly. We argue that in the presence of a persistently abnormal CTG trace, in a pregnancy with no clinical settings suggestive for either chronic or acute fetal hypoxemia, the absence of Wharton’s jelly should be taken into consideration in the differential diagnosis.

## Figures and Tables

**Figure 1 medicina-57-01268-f001:**
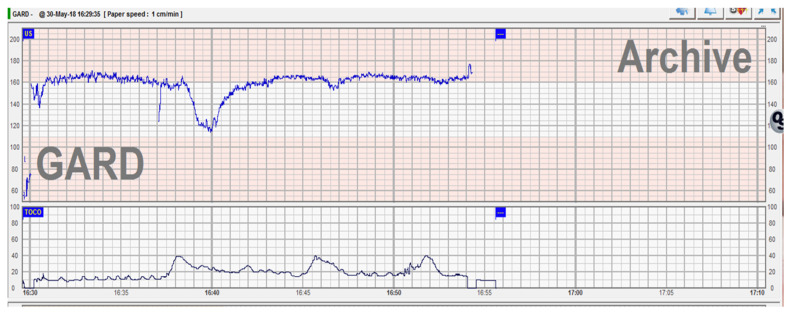
Abnormal CTG trace with tachycardia and late deceleration.

**Figure 2 medicina-57-01268-f002:**
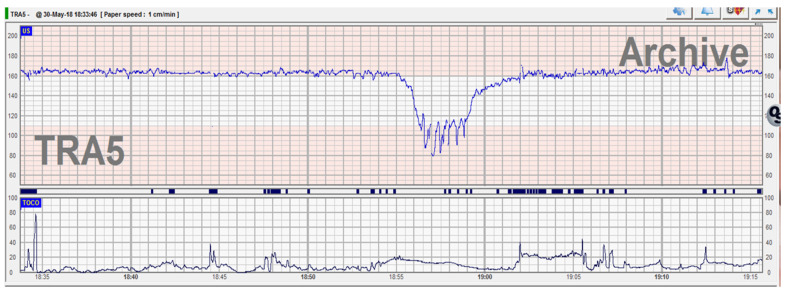
Abnormal CTG trace with 4 min prolonged deceleration at 2 h from the previous one.

**Figure 3 medicina-57-01268-f003:**
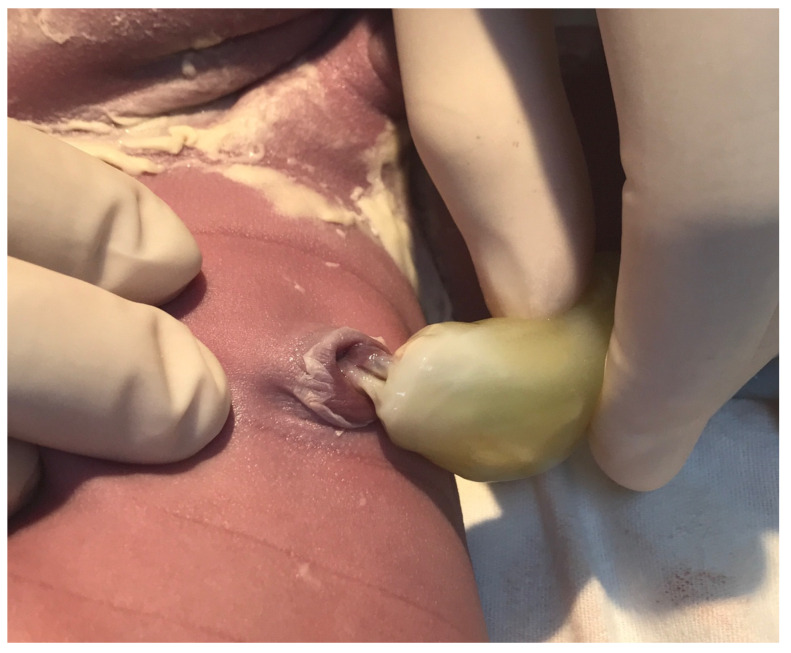
Absence of Wharton’s jelly at the umbilical cord insertion.

**Table 1 medicina-57-01268-t001:** Cases reported with an absence of Wharton’s jelly. MA—maternal age; GA—gestational age; wks.—weeks; wt.—weight; CTG—cardiotocography; AS—Apgar score; CS—cesarean section; g—grams.

Author	MA	GA wks.	Birth wt.	Mode of Delivery	CTG	Amniotic Fluid	AS	Fetal Outcome
Bergman et al. [9]	25	40 + 6	2270 g	Vaginal	Absent	Meconium	-	Stillborn
Labarrere et al. [10]	25	42	3220 g	CS	Variable and late deceleration	Meconium	2	Death after 2 h
Labarrere et al. [10]	30	40	4100 g	CS	Persistent late decelerations	Meconium	1	Death after 5 h
Labarrere et al. [10]	20	40	2920 g	Vaginal	Absent	Meconium	-	Stillborn
Kulkarni et al. [11]	19	38	2500 g	Vaginal	Variable	Clear	4	Alive with persistent vitellointestinal duct
Thomson et al. [12]	NK	NK	3450 g	CS	Not known	NK	0	Alive with severe impairment
Oliveira et al. [2]	22	NK	NK	Vaginal	Not known	NK	-	Stillborn
Damascenco et al. [13]	32	33 + 4	2260 g	Vaginal	Absent	NK	NK	Stillborn
Cole et al. [5]	31	40	3285 g	CS	Atypical variable decelerations	Meconium	6	Discharged after 4 days
Murphy et al. [14]	42	40 + 2	3680 g	CS	Persistent late decelerations	Meconium	8	Discharged after 14 days
Our case	29	39	3500	CS	Late decelerations after fetal movements	Clear	8	Discharged after 3 days

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
