# Peer review of "Absence of Wharton’s Jelly at the Abdominal Site of the Umbilical Cord Insertion. Rare Case Report and Review of the Literature"

_medicina, 2021, doi:10.3390/medicina57111268_

Round 1
Reviewer 1 Report
Introduction, 3rd paragraph line 41-43 " Complete gross and histopathologic .... need to reference this sentence
Case report , line 73 " was noted with no abnormal smell or aspect" No sure of the meaning of this sentence ? "aspect" sentence is confusing please re-write
For your literature search need to specify: search engines used, years of the search, search terms, and were there any exclusions for that search?
Author Response
Response to reviewers comments
Reviewer 1
Thank you for taking time to go through our manuscript. We really appreciate your feedback and feel that it has significantly improved this case report.
Please find below the answers to the raised questions. These answers are inserted in the manuscript marked in yellow.
Introduction, 3rd paragraph line 41-43 " Complete gross and histopathologic .... need to reference this sentence
Thank you, reference has been added.
Case report , line 73 " was noted with no abnormal smell or aspect" No sure of the meaning of this sentence ? "aspect" sentence is confusing please re-write
Thank you, we agree and have replaced the sentence.
For your literature search need to specify: search engines used, years of the search, search terms, and were there any exclusions for that search?
Thank you, we have now added the description of the search in the second paragraph of the Discussion section.

Reviewer 2 Report
You have reported a case of a fetus that had the absence of Wharton's jelly and survived. You have also described a literature review of previous case reports.
1. Before this manuscript is published, I have some concerns and questions.
My concerns and questions are described below.
2. Did you perform CTG on Antepartum? If so, please describe the findings.
I had an artificial rupture of membranes during delivery. I don't understand why my membranes broke. Please add an explanation.
3. Wouldn't it have been better for you to have shifted to a C-section at the timing of Figure 1?
4. Please explain the mechanism for the occurrence of the absence of Wharton's jelly in the Discussion section.
5. Are there any differences from previous case reports of the absence of Wharton's jelly? If there are any new findings, please describe them.
6. Is there a relationship between the absence of Wharton's jelly and excessive episodic fetal movements? Is there a relationship between the absence of Wharton's jelly and excessive episodic fetal movements? If there was no excessive episodic fetal movements and the patient was not hospitalized, more severe fetal hypoxia and fetal death could have been predicted.
7. The structure of the discussion section should be changed. It is difficult for readers to read.
Author Response
Reviewer 2
Thank you for taking time to go through our manuscript. We really appreciate your feedback and feel that it has significantly improved this case report.
Please find below the answers to the raised questions. These answers are inserted in the manuscript marked in yellow
You have reported a case of a fetus that had the absence of Wharton's jelly and survived. You have also described a literature review of previous case reports.
1. Before this manuscript is published, I have some concerns and questions.
My concerns and questions are described below.
2. Did you perform CTG on Antepartum? If so, please describe the findings.
Thank you, unfortunately we have no CTG performed before labor. The patient had a low risk pregnancy and no routine antepartum CTG was performed. She presented in early labor and these are the only traces we have.
I had an artificial rupture of membranes during delivery. I don't understand why my membranes broke. Please add an explanation.
Thank you. In our hospital we are using 2015 FIGO consensus guidelines on intrapartum fetal monitoring and Omniview Sisporto central monitoring system with real time CTG analysis therefore we detected an abnormal trace. According to guidelines, immediate action to correct reversible causes, additional methods to evaluate fetal oxygenation were required. Because the patient was asking for a vaginal delivery and chronic hypoxia needed to be excluded, artificial rupture of membranes was performed to rule-out the presence of meconium, possible infection or intraamniotic bleeding. Explanation was added to the manuscript.
- Wouldn't it have been better for you to have shifted to a C-section at the timing of Figure 1?
Thank you, yes in retrospect, however, the patient was keen on vaginal delivery and initially declined a caesarean section. Because active fetal movements were present, the plan agreed was for artificial rupture of membranes and reassessment at 2 hours under continuous fetal monitoring.
- Please explain the mechanism for the occurrence of the absence of Wharton's jelly in the Discussion section.
Thank you, we have now added this in the Discussion section.
- Are there any differences from previous case reports of the absence of Wharton's jelly? If there are any new findings, please describe them.
Thank you, we have now clearly specified the differences in our case from other cases reported in the literature. Our case comes in contradiction with previously reported cases where absence of Wharton's jelly was associated with either meconium-stained liquor [4] or oligohydramnios, fetal growth restriction, preterm delivery and perinatal death.
The main difference with previously reported cases was the association with active fetal movements.
- Is there a relationship between the absence of Wharton's jelly and excessive episodic fetal movements? Is there a relationship between the absence of Wharton's jelly and excessive episodic fetal movements? If there was no excessive episodic fetal movements and the patient was not hospitalized, more severe fetal hypoxia and fetal death could have been predicted.
Thank you, we do not think that there is a relationship between excessive fetal movement and the absence of Wharton’s jelly. We have now clarified this in the text.
- The structure of the discussion section should be changed. It is difficult for readers to read.
Thank you, we have now changed the structure of the Discussion section.

Round 2
Reviewer 2 Report
The authors responded to my requests in a precise manner. I recommend that this manuscript be accepted for publication.